# Combined Pulsed Electric Field with Antimicrobial Caps for Extending Shelf Life of Orange Juice

**Tony Z. Jin [1],\* and Ramadan M. Aboelhaggag [2]**

[1] Food Safety and Intervention Technologies Research Unit, Eastern Regional Research Center, Agricultural Research Service, U.S. Department of Agriculture, 600 E. Mermaid Lane, Wyndmoor, PA 19038, USA

[2] Food Technology Department, Food Industries and Nutrition Institute, National Research Center (NRC), 33 El Bohouth St., Dokki, Cairo 12622, Egypt

\* Correspondence: tony.jin@usda.gov; Tel.: +1-215-836-6904; Fax: +1-215-233-6445

**Abstract:** The purpose of this study was to investigate the effectiveness of combined pulsed electric fields (PEF) and antimicrobial packaging treatment in maintaining the quality and stability of orange juice stored at 10 °C. Orange juice was treated by PEF and stored in glass jars with antimicrobial caps coated with 10 μL of carvacrol essential oil (AP). Microbial reductions and physiochemical properties of juice samples were determined after treatments and during storage at 10 °C. Orange juice samples subjected to the combined treatment (PEF+AP) had the lowest yeast and mold populations after 14 day-storage at 10 °C. There were no significant differences in pH, acidity, color, total soluble solid contents, total phenol compounds, and Vitamin C among all samples after treatments. Storage studies showed that PEF, AP, and PEF+AP treatments maintained the quality and stability of orange juice stored at 10 °C for 5 weeks but lost Vitamin C. This study provides valuable information to juice processors for consideration and design of nonthermal pasteurization with antimicrobial packaging of juice products.

**Keywords:** PEF; antimicrobial packaging; combination; orange juice; shelf-life; quality

## 1. Introduction

Nonthermal food preservation technologies have been extensively studied worldwide. Compared with thermal processing, non-thermal processes offer several advantages including low processing temperatures, low energy utilization and the retention of flavors, nutrients, and a fresh-like taste of foods, while inactivating microbial populations. Therefore, in recent years, non-thermal processes have been considered as a potential technology to replace or complement the traditional thermal processing of foods. Two examples of nonthermal food preservation technologies are pulsed electric fields and antimicrobial packaging.

Pulsed electric field processing is a promising nonthermal food preservation technology for liquid foods. During PEF processing, food samples are placed between two electrodes and exposed to high voltage pulses with pulse widths from microseconds to milliseconds to form an electric field between the two electrodes. Several studies in the literature have demonstrated the ability of PEF treatment to stop spoilage and kill foodborne pathogenic microorganisms in various food products [1–7] without scarifying their nutritional and quality attributes [8–14]. Antimicrobial packaging uses food packaging materials incorporating antimicrobials. The application of antimicrobials into packaging materials (films or coatings) allows for the migration of the antimicrobial to the food surface in a controlled manner, therefore providing a continuous antimicrobial effect on the food during extended periods of storage. Jin et al. (2010) [15] incorporated sodium benzoate and potassium sorbate in polylactic acid films and compared their antimicrobial effects with direct addition of these antimicrobials to strawberry puree. Their results demonstrated that the controlled release of potassium sorbate and sodium benzoate from the films

allowed for a greater reduction in populations of *E. coli* O157:H7 artificially inoculated into strawberry puree and natural microflora compared to direct addition of the antimicrobials to the puree. Other researchers have also shown that the controlled release of antimicrobials from films exhibited prolonged antimicrobial activity compared with direct addition [16–18]. Most studies have worked on solid foods and a limited number of studies are available for liquid foods. Our previous studies also demonstrated that antimicrobials that were incorporated in pectin and polylactic acid composite films effectively inactivated populations of *E. coli* O157:H7, *Listeria monocytogenes*, and *Salmonella* in liquid egg albumen, milk, orange juice, and strawberry puree [15,19–21]. As packaging is the last step in food production, antimicrobial packaging provides a final defense against microbial contamination and recontamination, and quality deterioration. Therefore, the application of antimicrobial packaging is an effective approach to enhancing food safety, improving food quality, extending food shelf-life, and reducing food waste. Several studies have investigated the synergistic effects of using nonthermal processing techniques with antimicrobial or preservative compounds; these compounds are directly mixed in liquid food before nonthermal processing. Pina-Perez et al. (2012) [22] reported the synergistic effect of PEF and various concentrations of cinnamon against populations of *Salmonella typhimurium* artificially inoculated in skim milk. Their results demonstrated that the rate of inactivation increased with increased electrical field strength and increased concentrations of cinnamon. Specifically, *Salmonella* populations were reduced by 52% when the concentration of cinnamon was 5% and a PEF treatment of 10 kV/cm at 3000 μs was applied. McNamee et al. (2010) [23] investigated the synergistic effect of PEF and bacteriocins against populations of *Escherichia coli* K12 and *Listeria innocua* and spoilage bacteria in orange juice. Their results indicated that the combination of both treatments reduced populations of *E. coli* K12, *L. innocua*, and spoilage yeasts by more than 5 log. Several studies have demonstrated the synergistic effects of essential oils (EO) combined with PEF treatments against various foodborne pathogens and spoilage microorganisms in fruit juices [4,24,25]. Jin et al. (2014, 2021) [14,26], reported the effectiveness of PEF treatment and an antimicrobial bottle coating in extending the shelf-life of pomegranate juice and apple juice, respectively. However, the combination of PEF and antimicrobial packaging for juices and beverages have not been widely studied or reported. The results from a previous study conducted in our lab showed that the combination of antimicrobial packaging with PEF processing extended the shelf life of apple juice significantly [26]. Therefore, the objective of this study was to investigate the antimicrobial efficacy of these combinations against spoilage microorganisms in orange juice and evaluate their synergistic effects on the quality of the juice.

## 2. Materials and Methods

### 2.1. Preparation of Juice Samples

No-pulp orange juice was purchased from a local supermarket for the storage study. For the challenge study to test native microflora reduction, orange juice was placed at room temperature (approximately 25 °C) for two weeks until the yeast and mold (YMC) populations reached between $1 \times 10^5$ and $1 \times 10^6$ CFU/mL.

### 2.2. Treatments of Pulsed Electric Field

A pulsed electric field system (laboratory-scale, OSU-4H) was used in this study as described by Jin et al. (2021) [26]. Briefly, the PEF system consisted of a pulse generator (Model 9410, Quantum Composers, Inc., Bozeman, MT, USA) and six pairs of treatment chambers; the treated sample was cooled by passing through a cooling coil submerged in a water bath after passing through each pair of treatment chambers. The PEF treatment conditions were: 60 mL/min flow rate, 19 kV/cm field strengths, 1250 pulses per second, 2 μs pulse width, and 181 μs total treatment time. Figure 1 shows the setup of the laboratory-scale pulsed electric field system used in this study.

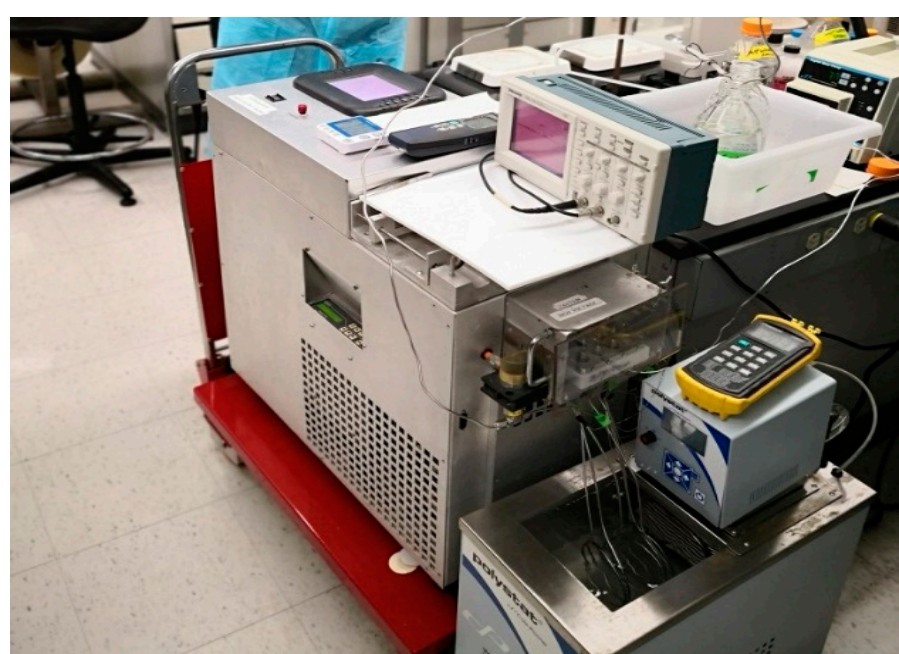

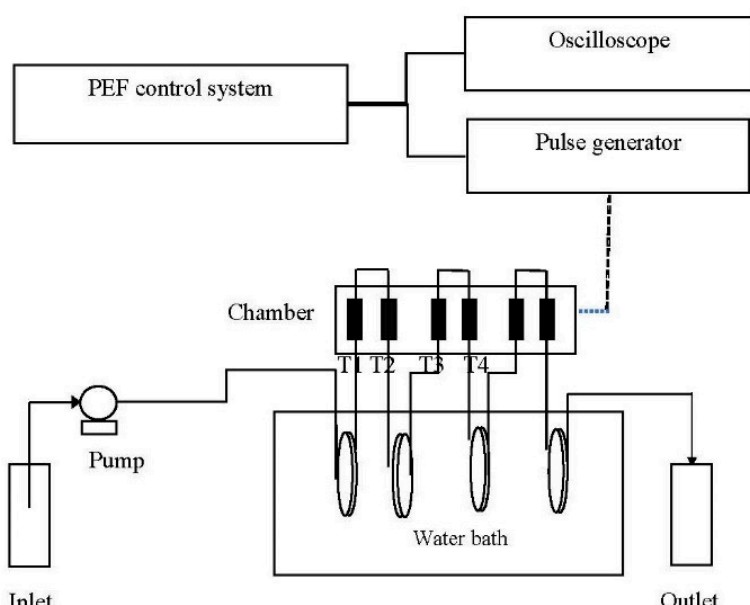

**Figure 1.** Photos of the OSU-4H PEF processing system (**top**) and flow chart (**bottom**). Dot line represents a pulse supply cable.

### 2.3. Antimicrobial Caps

Chitosan ((low molecular weight, 150 kDa, 75–85% deacetylation) and carvacrol essential oil (>98% purity) were purchased from Sigma–Aldrich (St. Louis, MO, USA). Lactic acid and acetic acid were purchased from Fisher Scientific (Fairlawn, NJ, USA). Five mg chitosan, 500 mg corn fiber gum (ERRC, PA, USA), and 100 μL carvacrol were added to a 10 mL acid solution consisting of 1% lactic acid and 1% acetic acid. The mixture was stirred using a magnetic stir plate at ambient temperature under constant agitation for 12 h to make an antimicrobial emulsion solution. A 1-mL aliquot of the solution was coated on the inner surface of the bottle cap and the coated cap was dried under a chemical hood overnight making it ready for use (Figure 2). The antimicrobial formulation and volume

applied on the caps were based our preliminary experiments and previous study [26], which showed that the caps had reasonable antimicrobial effect with least unpleasant odor.

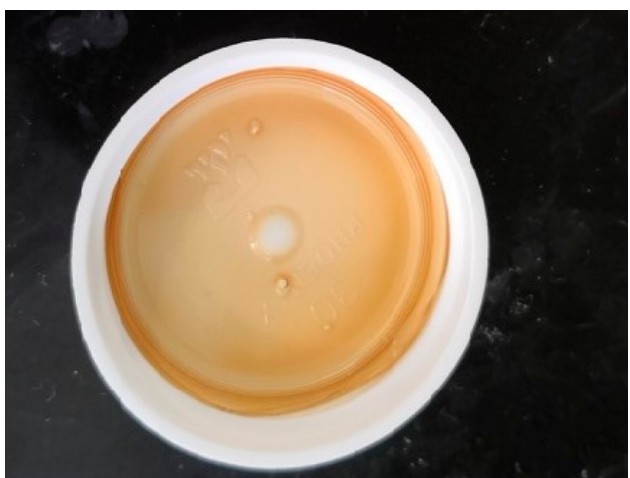 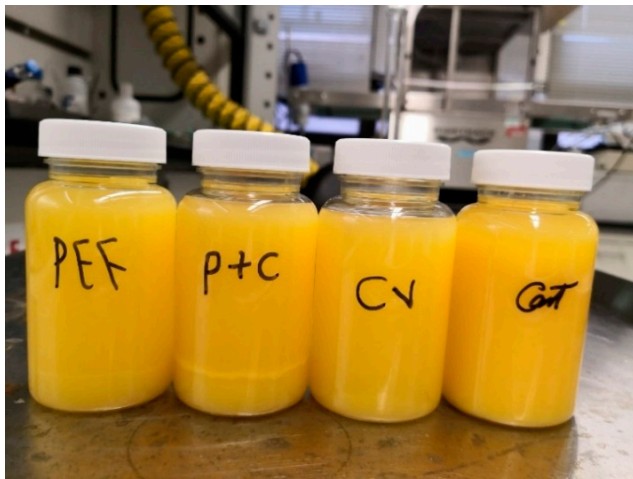

**Figure 2.** Photos of antimicrobial coated jar cap (**left**) and packaged orange juice (**right**).

Glass jars (250 mL) were used for the storage study. After PEF processing, each juice sample (200 mL) was poured into a glass jar and covered with a regular cap (not coating) or coated cap (AP) (Figure 2). All the glass jars were stored at 10 ± 1 °C until sampling for analyses.

### 2.4. Microbiological Analysis

After processing or storage, appropriate dilutions of the juice samples were made in Butterfield's phosphate buffer (Hardy Diagnostics, Santa Maria, CA, USA). One hundred µL of the appropriate dilutions were surface-plated in duplicate onto Plate Count Agar (PCA, BBL/Difco Laboratories, Sparks, MD, USA) plates to determine surviving total aerobic bacteria (TAB) populations, and onto Dichloran Rose Bengal Chloramphenicol Agar (DRBC, Merck, Germany) plates to determine surviving yeast and mold (YMC) populations. The PCA plates were incubated at 37 °C for 24 h, and the DRBC plates were incubated at 25 °C for 5 days before counting colony forming units (CFU). Colonies were then counted and expressed as $\log_{10}$ colony forming units per milliliter of juice ($\log_{10}$ CFU/mL). All samples were analyzed in duplicate, and two replicates of each dilution were prepared and plated.

### 2.5. Determinations of pH, Titratable Acidity, and Total Soluble Solids

A digital pH-meter (Thermo Electron Corp., Beverly, MA, USA) was used to measure the pH values of the juice samples and titratable acidity (TA) was determined using the method described by Bhat et al. (2011) [27]; the results were expressed as milligrams of citric acid per 100 mL juice. Total soluble solids (TSS) were measured using a digital refractometer (Reichert, Inc., Depew, NY, USA) expressed as °Brix. All measurements were performed at approximately 25 °C.

### 2.6. Determination of Color

Color measurements of the juice samples were performed at 25 °C using a portable colorimeter (JZ-300, Shengzhen Kingwell Instrument Co. Ltd., Guangdong, China). Color values were expressed as *L*\* (lightness or brightness/darkness), *a*\* (redness/greenness) and *b*\* (yellowness/blueness). Three measurements for each sample were conducted and averaged.

## 2.7. Determination of Ascorbic Acid (Vitamin C) Content

The ascorbic acid in the tested juice samples was determined by using the 2,6-dichlorophenol indophenol (DCPIP) visual titration method described by Ranganna (2001) [28]. Five mL of juice sample was used. The obtained results were expressed as mg ascorbic acid per 100 mL juice sample. Three measurements for each sample were conducted and averaged.

## 2.8. Determination of Total Phenolics Content

Total phenolic concentration was estimated using the Folin–Ciocalteu method described by Kwaw et al. (2018) [29]. The total phenolic concentration was expressed in terms of micrograms of gallic acid equivalent per milliliter of juice.

## 2.9. Statistical Analysis

At least three independent trials were conducted for all the treatments. Colony counts were converted to log CFU/mL before SAS analysis. SAS Version 9.2 (SAS Institute, Inc., Cary, NC, USA) software was used to analyze all data, and significant differences ($p < 0.05$) between control and treated samples were determined by Fisher's Least Significant Difference Test (LSD).

## 3. Results and Discussion

### 3.1. Effect of Treatments on Microbial Reductions

The PEF and antimicrobial packaging (AP) were investigated individually or as a combined treatment for their antimicrobial efficacy against TAB and YMC in juice samples. Because the initial populations of TAB and YMC in juice samples from the supermarket were very low, only small reductions were obtained by PEF, AP, and PEF+AP treatments. Throughout the five-week storage period at 10 °C, bacterial and fungal populations increased in the control samples by 0.5 log CFU. Bacterial and fungal populations in all the other samples increased by 1 log CFU. As a result, juice samples whose YMC and TAB counts were 5–6 log CFU and 2–3 log CFU, respectively, were used for further studies. Figure 3 displays the changes in populations of TAB (A) and YMC (B) after treatments and during storage. The TAB populations were reduced by PEF and PEF+AP right after the treatments (Day 0). The TAB populations were further reduced by all three treatments (PEF, AP, and PEF+AP) on Day 1 and were less than 1 log CFU (detection limit) in all samples after three days (Figure 3A). The PEF and PEF+AP reduced YMC populations from 5.5 log CFU to 1.2 log CFU on Day 0. The YMC populations increased in PEF treated samples to 3.8 log CFU on day 14, while the YMC populations in PEF+AP treated samples decreased to less than 1 log CFU (Figure 3B). In a separate study, inoculated *E. coli* K12 did not survive in orange juice after 7 days storage at 10 °C.

The addition of antimicrobial packaging (AP) to PEF treatment did not significantly contribute to the inactivation rate on Day 0. This was because the antimicrobial agent was slowly released from the cap coating to the headspace of the glass jars, generating gradual but continuous antimicrobial effects. As the antimicrobial effect from the cap increased during storage, the combined PEF+AP enhanced the microbial reductions with additive biocidal effect. Further studies need to be conducted to determine if increased carvacrol concentration in the coating can enhance its antimicrobial effect. Jin et al. (2021) [26] investigated the effectiveness of PEF and AP, either individually or as a combined treatment, on the inactivation of *E. coli* K12 populations artificially inoculated in apple juice stored at 10 °C. Their results indicated that the combination of PEF treatment at a lower electrical field strength with antimicrobial packaging (PEF+AP) achieved similar reductions in microbial populations to those obtained using PEF treatment alone at a higher electrical field strength.

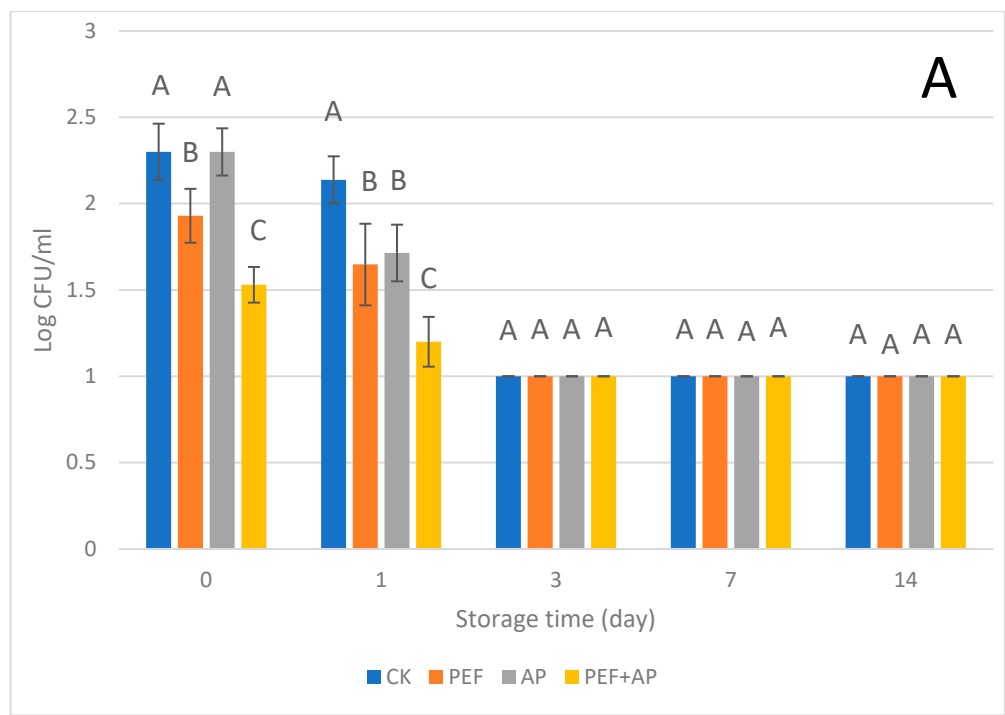

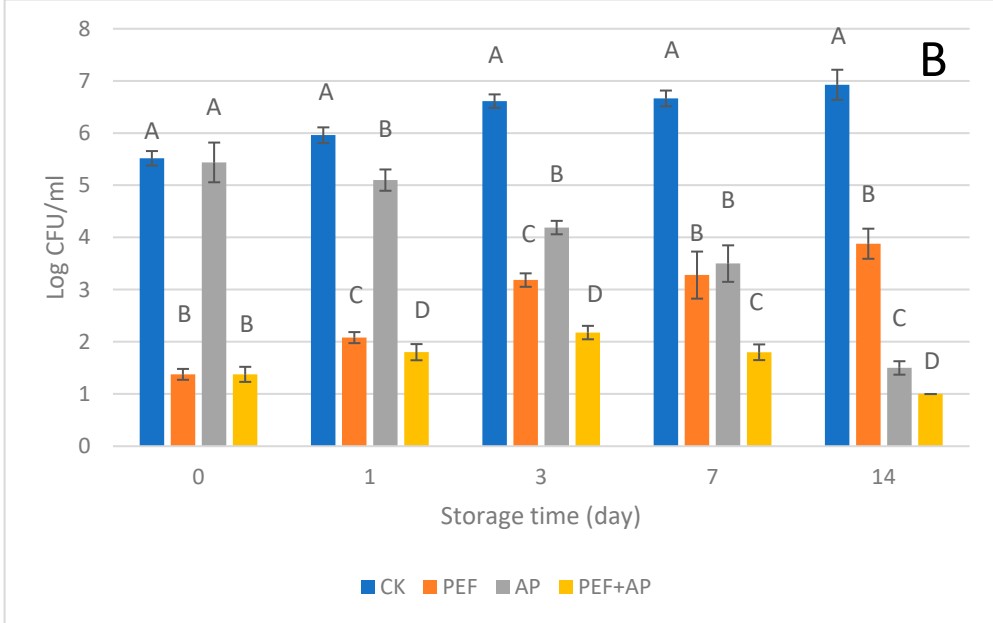

**Figure 3.** Survival of total aerobic bacteria (TAB) (**A**) and yeast and mold (YMC) (**B**) in orange juice after treatments and during storage at 10 °C. CK: control; PEF: PEF field strength of 19 kV/cm; AP: juice samples were packaged in a glass jar with coated cap; PEF+AP: PEF followed by AP. Bars with different letters on the same day are significantly different ($p < 0.05$). Bars on 1 log CFU/mL represent the values below the detection limit (<1 log CFU/mL).

Yeasts and molds were the main cause of spoilage of highly acidic fruit juices, as they are capable of growth at pH 1.5 and at water activity values below 0.89 [30]. The results shown in Figure 3B indicate that all samples had increased YMC populations after 3 days.

The application of PEF to bacterial cells causes the formation of pores in the cell membrane, increasing its permeability. This allows for acidic molecules to enter the cytoplasm easily, allowing for a decrease in the intracellular pH and eventually causing cell death [31]. Therefore, acidic foods are of little concern considering the occurrence

of sublethal injury in bacteria. In contrast, natural microflora (YMC) in juice were more resistant to low pH because they have already adapted to the acidic environment. This explains the growth of YMC in control samples during storage. Jin et al. (2014) [14] demonstrated that yeasts and molds in pomegranate juice had lower rates of injury than artificially inoculated populations of *E. coli* ATCC 35218 when subjected to combined PEF treatment and antimicrobial bottle coating; the rates of injury were 80% and 92.8%, respectively. However, the results in Figure 3B indicate that the addition of AP to PEF treatments (PEF+AP) significantly lowered the survival of YMC in juice samples after storage for 7d. The results suggest that the use of AP in combination with PEF and PL is an additional approach to prevent or delay spoilage during storage for long periods of time. Several studies have focused on the application of a combined PEF treatment and antimicrobials such as nisin and sorbic acid and demonstrated that these combinations limited the recovery capability of damaged cells and thus increased the lethality of PEF [32–34].

### 3.2. Effect of Treatments on Quality Characteristics

The changes in quality and nutritional attributes of orange juice after PEF, AP, and combined treatments during storage at 10 °C are presented in Table 1.

**Table 1.** Quality and nutrition changes in orange juice during storage at 10 °C.

| | | Storage (Weeks) | | | | | |
|---|---|---|---|---|---|---|---|
| | | 0 | 1 | 2 | 3 | 4 | 5 |
| pH | Control | 3.91 ± 0.01 a | 3.93 ± 0.01 a | 3.80 ± 0.02 a | 3.88 ± 0.01 a | 3.91 ± 0.01 a | 3.76 ± 0.01 a |
| | PEF | 3.90 ± 0.01 a | 3.89 ± 0.01 b | 3.79 ± 0.02 a | 3.88 ± 0.01 a | 3.90 ± 0.01 a | 3.76 ± 0.01 a |
| | AP | 3.91 ± 0.01 a | 3.87 ± 0.01 b | 3.78 ± 0.02 a | 3.88 ± 0.01 a | 3.90 ± 0.01 a | 3.76 ± 0.01 a |
| | PEF+AP | 3.90 ± 0.01 a | 3.88 ± 0.01 b | 3.78 ± 0.02 a | 3.88 ± 0.01 a | 3.90 ± 0.01 a | 3.76 ± 0.01 a |
| Acidity (mg/mL) | Control | 3.15 ± 0.05 a | 3.14 ± 0.03 b | 3.49 ± 0.02 a | 3.53 ± 0.03 a | 3.51 ± 0.02 a | 3.76 ± 0.02 a |
| | PEF | 3.18 ± 0.06 a | 3.23 ± 0.03 a | 3.48 ± 0.04 a | 3.50 ± 0.02 a | 3.53 ± 0.03 a | 3.68 ± 0.02 b |
| | AP | 3.15 ± 0.05 a | 3.21 ± 0.03 a | 3.51 ± 0.02 a | 3.51 ± 0.04 a | 3.51 ± 0.01 a | 3.71 ± 0.02 b |
| | PEF+AP | 3.18 ± 0.06 a | 3.21 ± 0.02 a | 3.49 ± 0.02 a | 3.55 ± 0.02 a | 3.50 ± 0.01 a | 3.69 ± 0.02 b |
| TSS | Control | 11.23 ± 0.06 a | 11.03 ± 0.06 a | 10.93 ± 0.12 a | 11.17 ± 0.06 a | 11.00 ± 0.01 a | 11.03 ± 0.06 a |
| | PEF | 11.10 ± 0.11 a | 10.90 ± 0.10 a | 11.07 ± 0.12 a | 11.27 ± 0.12 a | 10.97 ± 0.06 a | 10.90 ± 0.01 a |
| | AP | 11.23 ± 0.06 a | 11.10 ± 0.10 a | 11.17 ± 0.06 a | 11.33 ± 0.12 a | 11.10 ± 0.10 a | 11.00 ± 0.01 a |
| | PEF+AP | 11.10 ± 0.10 a | 11.13 ± 0.12 a | 11.13 ± 0.12 a | 11.30 ± 0.02 a | 11.10 ± 0.01 a | 11.17 ± 0.07 a |
| Color *L** value | Control | 31.17 ± 0.31 a | 29.87 ± 0.55 a | 28.53 ± 0.67 a | 29.13 ± 0.45 a | 28.80 ± 0.26 a | 28.47 ± 0.21 a |
| | PEF | 30.20 ± 0.56 a | 29.80 ± 0.53 a | 29.97 ± 0.93 a | 29.13 ± 0.15 a | 28.57 ± 0.31 a | 28.10 ± 0.10 a |
| | AP | 31.17 ± 0.32 a | 28.37 ± 0.15 a | 27.97 ± 0.23 a | 28.37 ± 0.61 a | 28.57 ± 0.38 a | 28.53 ± 0.23 a |
| | PEF+AP | 30.20 ± 0.45 a | 30.27 ± 0.25 a | 29.47 ± 1.21 a | 28.43 ± 0.49 a | 29.07 ± 0.15 a | 27.97 ± 0.15 a |
| Color *a** value | Control | 14.43 ± 0.51 a | 14.60 ± 1.01 a | 14.30 ± 0.36 a | 15.67 ± 0.85 a | 14.97 ± 0.31 a | 16.40 ± 0.60 a |
| | PEF | 15.97 ± 0.41 a | 16.33 ± 0.42 a | 13.73 ± 1.10 a | 15.23 ± 1.33 a | 14.17 ± 0.90 a | 15.30 ± 0.46 b |
| | AP | 14.43 ± 0.55 a | 15.17 ± 0.25 a | 13.77 ± 1.84 a | 13.80 ± 0.26 b | 13.97 ± 0.50 a | 15.13 ± 0.49 b |
| | PEF+AP | 15.97 ± 0.46 a | 14.93 ± 0.25 a | 14.73 ± 0.55 a | 13.93 ± 0.12 b | 14.60 ± 0.36 a | 15.60 ± 0.87 b |
| Color *b** value | Control | 4.67 ± 0.35 a | 4.27 ± 0.21 a | 4.33 ± 0.31 b | 3.73 ± 0.06 a | 4.53 ± 0.35 a | 3.73 ± 0.12 a |
| | PEF | 3.97 ± 0.25 a | 3.53 ± 0.31 b | 4.57 ± 1.04 b | 3.83 ± 0.70 a | 4.17 ± 0.35 a | 3.97 ± 0.15 a |
| | AP | 4.67 ± 0.24 a | 3.47 ± 0.15 b | 4.23 ± 1.05 b | 3.83 ± 0.35 a | 4.37 ± 0.38 a | 4.03 ± 0.61 a |
| | PEF+AP | 3.97 ± 0.33 a | 3.93 ± 0.25 a | 5.10 ± 0.30 a | 4.20 ± 0.56 a | 4.57 ± 0.21 a | 4.00 ± 0.20 a |

**Table 1.** *Cont.*

|  |  | Storage (Weeks) | | | | | |
|---|---|---|---|---|---|---|---|
|  |  | 0 | 1 | 2 | 3 | 4 | 5 |
| Total phenol (μg/100 μL) | Control | 75.47 ± 3.21 a | 65.59 ± 2.88 a | 77.21 ± 3.32 b | 74.97 ± 1.55 a | 66.32 ± 3.08 b | 66.15 ± 1.65 a |
|  | PEF | 73.55 ± 3.25 a | 65.71 ± 2.97 a | 76.53 ± 3.44 b | 73.55 ± 2.12 a | 67.03 ± 2.79 b | 67.74 ± 2.33 a |
|  | AP | 75.47 ± 3.33 a | 65.32 ± 1.99 a | 78.01 ± 1.09 b | 75.11 ± 2.33 a | 68.89 ± 2.55 b | 68.53 ± 1.12 a |
|  | PEF+AP | 73.55 ± 3.09 a | 66.30 ± 2.03 a | 88.28 ± 1.77 a | 75.12 ± 2.55 a | 72.34 ± 2.12 a | 69.24 ± 1.34 a |
| Vitamin C (mg/100 mL) | Control | 34.44 ± 1.12 a | 26.83 ± 1.23 a | 16.47 ± 1.77 b | 12.58 ± 1.22 b | 5.91 ± 0.77 b | 0.83 ± 0.05 b |
|  | PEF | 33.98 ± 1.09 a | 26.86 ± 0.98 a | 22.58 ± 1.45 a | 16.86 ± 2.09 a | 16.86 ± 1.13 a | 5.55 ± 0.30 a |
|  | AP | 34.44 ± 1.11 a | 27.30 ± 1.21 a | 15.83 ± 0.79 b | 16.66 ± 1.39 a | 4.25 ± 0.59 b | 0.92 ± 0.12 b |
|  | PEF+AP | 33.98 ± 1.03 a | 27.11 ± 1.07 a | 22.41 ± 1.10 a | 8.52 ± 1.32 c | 2.97 ± 0.32 c | 1.02 ± 0.16 b |

PEF: PEF field strength of 19 kV/cm; AP and PEF+AP: juice samples were packaged in a glass jar with antimicrobial coated cap. Means with the same letter in the same column are not significantly different ($p < 0.05$).

No significant differences ($p > 0.05$) were found in pH, acidity, and TSS among all treatments. Similar results were previously reported in apple juice [35–38], mulberry juice [29], apple and cranberry juice blend [8], and pomegranate juice [39].

Color is an important quality parameter used to determine the juice freshness. Only the control samples had slightly higher redness (*a\**) and lower yellowness (*b\**) values after storage. The PEF treated samples did not have significant color changes (Table 1). Numerous studies have demonstrated that PEF treatment could maintain the color of juice throughout storage, even when the field strength and treatment time were increased [11,14,40].

Generally, not all attributes were significantly affected by the treatments. Although the results reveal that the total phenolic concentration in the treated juice samples remained relatively stable throughout storage, the contents of vitamin C in juice samples were significantly affected and reduced during storage, regardless of the treatment they were subjected to. Tran et al. (2004) [41] also observed that storage of orange juice for long periods of time could cause Vitamin C loss. The PEF treated juice samples had the least loss of vitamin C during storage compared with other samples. Similar results were observed when apple juice was subjected to the same treatment conditions. This study further proved that storage temperature is a major factor that can influence Vitamin C levels; therefore, juices should be stored at a lower temperature (<10 °C).

The effect of pulsed electric fields on microbial inactivation is due to the formation of pores in the cell membrane [42]. Applied high voltage pulses induce electroporation of cell membranes in microbes, plant, and animal cells [43,44]. The electroporation forms pores in the cell membrane, causing the movement of intracellular as well as extracellular molecules across the cell membrane, consequently, disrupting cellular homeostasis and eventually leading to cell death [45]. Because PEF processing only generates a slight increase in temperature (<10 °C), the thermal impact by PEF on the quality of juice products is negligible.

Many essential oils have been used as gaseous antimicrobials. However, the unpleasant odor is a concern. In our study, we used carvacrol, one of the relative mild flavor oils, and relative low concentration (100 μL in 250 mL juice). Therefore, the unpleasant odor had disappeared after 1 week's storage. A formal sensory study for the juice taste will be conducted in the near future.

To the best of our knowledge, this is the second study that investigates the combined treatments of PEF and AP (antimicrobial released from cap coat) on the inactivation of spoilage microorganisms in juices and on the qualitative and nutritional attributes of juice. This study has not explored the optimal treatment conditions for PEF parameters (field strength, pulse width, frequency, total treatment time), and other cap coats with different kinds and concentrations of essential oils. Future studies should focus on the optimal treatment conditions needed to enhance the effectiveness of the combined PEF+AP treatment.

## 4. Conclusions

The results from this study indicated that the combined PEF+AP treatment was effective in reducing microbial levels in juices without negatively impacting their physico-chemical properties and nutritional attributes. Five log reduction of foodborne pathogens is required by the US regulation for juice pasteurization. To preserve the quality and nutritional properties of juice products, use of nonthermal processes is a better choice. However, for most nonthermal processes, a single treatment may not obtain such a microbial reduction. Alternatively, multiple (i.e., hurdle) treatments can be applied, which are permitted if juices are processed in the same facility [46]. Therefore, as a hurdle technique, the integration of PEF and AP treatments into juice processing lines, provides an alternative approach to meet this FDA mandatory requirement and improve the safety, quality, and storage stability of juices.

**Author Contributions:** Conceptualization, methodology, investigation, project administration, supervision, T.Z.J.; experiments and data collection, R.M.A. All authors have read and agreed to the published version of the manuscript.

**Funding:** This study was also funded by the USDA-ARS CRIS project 8072-41420-026-00D through ARS National Program 108.

**Data Availability Statement:** Data are available from the authors upon request.

**Acknowledgments:** The authors thank Anita Parameswaran for excellent technical assistance. Author Aboelhaggag thanks the Science and Technology Development Fund (STDF) of Egypt for providing the fellowship to work at ERRC-ARS-USDA. All work was done at ERRC-ARS-USDA. Mention of trade names or commercial products in this publication is solely for the purpose of providing specific information and does not imply recommendation or endorsement by the U.S. Department of Agriculture. USDA is an equal opportunity provider and employer.

**Conflicts of Interest:** The authors declare no conflict of interest.

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
