# Peer review of "Combined Pulsed Electric Field with Antimicrobial Caps for Extending Shelf Life of Orange Juice"

_beverages, doi:10.3390/beverages8040072_

Round 1
Reviewer 1 Report
This work investigates the effectiveness of the combination of pulsed electric fields (PEF) and antimicrobial packaging treatment in maintaining the quality and stability of stored orange juice. Although there are many papers in the literature on the inactivation of microorganisms in liquid foods, few of them combine antimicrobial packaging. There are data on the use of nystatin and other chemical components, while the use of essential oil with antimicrobial activity in combination with non-thermal techniques is new. I am interested to know if the authors can include sensory analysis in their work? Essential oils are known to have a strong odor. Do they affect the taste of orange juice after storage?
Author Response
Thank you very much for your comments and suggestions.
We totally agree that EO might affect the sensory quality. Although we did not conduct formative taste study, we did smell tests when the samples opened for physicochemical analyses. We found that the EO was sensed only for the first week's samples, after then, no smell was detected. A formative taste study may be needed in the future study.
Reviewer 2 Report
Comments to Authors,
The manuscript entitled "Combined pulsed electric field with antimicrobial packaging for extending shelf life of orange juice" is suitable for publication in Beverages; however, it has to be improved in some aspects, considering the following specific comments.
Specific Comments:
Line 14: Separate the magnitude (quantity) from the measurement unit (°C). It should read as: at 10 °C.
Line 15: Should read as: … with 10 mL of …
Line 16: Separate the magnitude (quantity) from the measurement unit (°C). It should read as: at 10 °C.
Line 18: Separate the magnitude (quantity) from the measurement unit (°C). It should read as: at 10 °C.
Line 21: Apple juice? Please clarify!
Line 21: Should read as: … at 10 °C for 5 wk but …
Line 91: Should read as: … temperature (approximately 25 °C) for two wk …
Line 92: Should read as: … reached 1 ´ 105 to 1 ´ 106 CFU/mL.
Line 99: Should read as: 60 mL/min flow rate,
Line 107: It is recommended not to start the text with a number. It should read as: Five hundred mg chitosan… What was the molecular weight and degree of deacetylation of the chitosan used?
Line 108: Should read as: … and 100 mL carvacrol …
Line 109: Should read as: … to a 10 mL acid …
Line 111: It should read as: … for 12 h to make …
Line 112: Should read as: A 1-mL aliquot …
Line 115: It should read as: Two hundred fifty mL … , 200 mL of
Line 117: Should read as: … at 10 ± 1 °C until …
Line 124: It should read as: One thousand mL of the …
Line 128: Should read as: … at 37 °C for …
Line 129: Should read as: … at 25 °C for 5 d before …
Line 130: It should read as: … as log10 colony …
Line 131: It should read as: … of juice (log10 CFU/mL).
Line 136: It should read as: Ten mL of …
Line 138: It should read as: … per 100 mL juice.
Line 140: Should read as: … temperature (approximately 25 °C).
Line 142: Should read as: … at 25 °C using …
Line 147: Should read as: 2.7. Determination of ascorbic acid (vitamin C) content
Line 149: Should read as: Five mL
Line 150: Should read as: … juice sample (5 mL) was …
Line 154: It should read as: … per 100 mL juice …
Line 158: Should read as: A 2 mL freshly …
Line 159: It should read as: … to 0.2 mL of juice …
Line 160: Should read as: … at 25 °C for …
Line 166: Should read as: … to log CFU/mL before …
Line 176: Should read as: … at 10 °C,
Line 233: It should read as: … at 10 °C.
Table 1: Why are there no letters of statistical significance in this Table? Please indicate them to clarify this in the different variables analyzed between the different treatments!
Author Response
Thank you very much for your comments and suggestions.
We have insert a space before C, change ml to mL, and others per your suggestions. We have also added letters of statistical significance in this Table.